# Clinical evaluation of postoperative analgesia, cardiorespiratory parameters and changes in liver and renal function tests of paracetamol compared to meloxicam and carprofen in dogs undergoing ovariohysterectomy

Ismael Hernández-Avalos[1], Alexander Valverde[2], José Antonio Ibancovichi-Camarillo[3]*, Pedro Sánchez-Aparicio[4], Sergio Recillas-Morales[4], Jorge Osorio-Avalos[5], Desiderio Rodríguez-Velázquez[6], Agatha Elisa Miranda-Cortés[7]

1 Faculty of Veterinary Medicine, Universida Nacional Autónoma de Mexico, FES Cuatitlan, Guelph, Ontario, Canada, 2 Department of Clinical Studies, Ontario Veterinary College, University of Guelph, Guelph, Ontario, Canada, 3 Department of Veterinary Anesthesia and Analgesia, Faculty of Veterinary Medicine, Universidad Autónoma del Estado de México, Toluca, México, 4 Department of Pharmacology, Faculty of Veterinary Medicine, Universidad Autónoma del Estado de México, Toluca, México, 5 Department of Biostatistics, Faculty of Veterinary Medicine, Universidad Autónoma del Estado de México, Toluca, México, 6 Department of Surgery Education, Faculty of Veterinary Medicine, Universidad Autónoma del Estado de México, Toluca, México, 7 Department of Pharmacology and Veterinary Therapeutics, School of Higher Studies Cuautitlán, Universidad Nacional Autónoma de México, México, México

* ibanvet@gmail.com

## Abstract

### Background

In veterinary medicine, the administration of nonsteroidal anti-inflammatory analgesics (NSAIDs) for the control of postsurgical pain in dogs and cats is common given the anti-inflammatory, analgesic, and antipyretic effects of these drugs. This study compared the serum biochemical changes and postoperative analgesic effects of paracetamol, meloxicam, and carprofen in bitches submitted to an ovariohysterectomy using the Dynamic Interactive Visual Analog Scale (DIVAS) and Pain Scale of the University of Melbourne (UMPS) scoring systems.

### Methods

Thirty bitches of different breeds underwent elective ovariohysterectomies and were randomly assigned to one of three treatment groups: a paracetamol group [15 mg kg$^{-1}$ intravenous (IV)], a carprofen group (4 mg kg$^{-1}$ IV), and a meloxicam group (0.2 mg kg$^{-1}$ IV). All treatments were administered 30 minutes prior to surgery. Paracetamol was administered every 8 hours postoperatively for 48 hours total, while carprofen and meloxicam were intravenously administered every 24 hours. An evaluation of post-surgical pain was done with the DIVAS and the UMPS. The first post-surgical pain measurement was performed 1 hour after surgery and then 2, 4, 6, 8, 12, 16, 20, 24, 36, and 48 hours after surgery.

**Data Availability Statement:** All relevant data are within the manuscript and its Supporting Information files.

**Funding:** The Mexican College of Veterinary Anesthesiology and Analgesia provided some materials used in the study but did not play any role in study design, data collection and analysis, decision to publish, or preparation of the manuscript.

**Competing interests:** The authors have declared that no competing interests exist.

## Results

All groups exhibited a gradual reduction in pain throughout the postoperative period in both scales; however, neither scale significantly differed between the three treatment groups ($P > 0.05$) during the 48 postoperative hours.

## Conclusions

Paracetamol was as effective as meloxicam and carprofen for post-surgical analgesia in bitches subjected to elective ovariohysterectomy. The present study demonstrates that paracetamol may be considered a tool for the effective treatment of acute perioperative pain in dogs. Furthermore, this drug led to no adverse reactions or changes in the parameters assessed in the present study, indicating its safety.

## Introduction

The International Association for the Study of Pain (IASP) has defined pain as an "unpleasant sensory or emotional experience associated to a real or potential damage in a tissue, or that is described in terms of said damage" [1]. Acute or chronic pain that is not properly treated causes unnecessary suffering as it predisposes patients to medical complications and significantly increases hospitalisation and recovery times [2].

In veterinary medicine, the administration of nonsteroidal anti-inflammatory analgesics (NSAIDs) for the control of postsurgical pain in dogs and cats is common given the anti-inflammatory, analgesic, and antipyretic effects of these drugs [3]. NSAIDs represent a group of analgesics with a great structural diversity but which converge in a similar mechanism of action, wherein cyclooxygenase (COX) is inhibited. The COX enzyme is present in most tissue types and two forms have been identified: COX-1 and COX-2 [4], where subsequent studies have shown that both isoforms are constitutive and inducible [5,6]. A recently described third isoform, COX-3, has also been identified in the canine cerebral cortex, with minimal amounts found peripherally. COX-3 is thought to be inhibited by paracetamol [7,8]; however, its activity and physiological effects in dogs, rats, and humans have been the source of some debate and speculation [9,10].

Non-selective NSAIDs have a great number of side effects, including renal failure in the presence of hypotension, which may restrict their use in anesthetised hypotensive patients [3]. However, this view has changed with the development of more selective NSAIDs that appear to be safer in patients with kidney disease. This improved safety is likely associated with a greater specificity of the COX-2 isoenzyme [11], as supported by several veterinary studies suggesting that the perioperative use of carprofen and meloxicam, COX-2-preferential NSAIDs, is preferred over those drugs that do not show this selectivity [12–14].

Paracetamol has analgesic and antipyretic properties, but unlike NSAIDs, has no anti-inflammatory activity [15]. The mechanisms of action of this drug are diverse and among those that have been described with the inhibition the COX-3 isoenzyme [7], the indirect activation of CB1 cannabinoid receptors [16,17], and the inhibition the serotonergic descending pathway [18]. Paracetamol may also interact with opioidergic systems [19] or nitric oxide pathways [20]. Recently, the antiarrhythmic effects of paracetamol on the myocardium of dogs have also been described, as this drug reduces the activity of myeloperoxidase, which in turn significantly reduces the oxidation of low-density lipoproteins (LDLs) in macrophages. The

cardioprotective effects of paracetamol include its reduction of infarct sizes and mortality within 48 hours after this obstructive event, actions that imply a mechanism mediated by catalase/superoxide dismutase in a dog model [21,22]. The anti-inflammatory effects of this drug have also been explored in a placebo-controlled cross-over trial in dogs that underwent surgery of the third metacarpus of the thoracic limb. A daily dose of 1.5 g paracetamol (500 mg every 8 hours) modulated the acute postoperative inflammation reaction in these animals without leading to any clinical signs of adverse effects. This result revealed that paracetamol may prevent some post-operative or post-traumatic surgical sequelae in dogs [23].

Further evidencing its potential for post-surgical use, the administration of the recommended dose of paracetamol in dogs (10–15 mg $kg^{-1}$ every 8–12 hours) does not lead to gastrointestinal, renal, or platelet-related side effects [24]. Furthermore, its toxic effects are only observed in dogs when a dose in excess of 100 mg $kg^{-1}$ is used; whereas in cats, even a recommended dose of paracetamol can lead to toxicity due to glucoronyl transferase deficiency [25–30].

There is significant evidence, however, that paracetamol is safe when prescribed at a therapeutic dose and for a limited period of time in dogs [31]. In part, the varying physiological interactions that paracetamol has may be in part to do with its potentiation of the analgesic effects of morphine [32] and tramadol in the management of postoperative pain in humans. This interaction likely varies with veterinary application as the use of analgesic drugs to reduce the minimum alveolar concentration of inhaled anaesthetics has not been proven [33]. However, clinically, the analgesic efficacy of the combined use of hydrocodone and paracetamol leads to similar responses to tramadol as does a single agent for the treatment of postoperative pain in dogs undergoing tibial plateau levelling osteotomy [34].

Paracetamol can be used when other NSAIDs are contraindicated or may lead to intolerable side effects [8]. Paracetamol is included in all levels of the therapeutic pain ladder of the World Health Organization [17] and is recommended for first line use by the American Society of Anesthesiologists in multimodal analgesic regimens for the management of postoperative pain [35]. However, in animals, paracetamol's use is more infrequent and limited (in time and amount) given its potential for toxicity even at therapeutic doses [30]. The aim of the present study was thus to compare serum biochemical changes and the postoperative analgesic effects of paracetamol, meloxicam, and carprofen in bitches post-ovariohysterectomy using the DIVAS and UMPS pain assessment scales. Our hypothesis was that paracetamol would provide analgesic effects, similar to those produced by meloxicam and carprofen, without producing any changes in the assessed parameters, indicating its safety.

## Materials and methods

### Animals

Thirty client-owned bitches of a variety of breeds (11 mixed breed, 7 poodle, 3 beagle, 2 schnauzer, 2 pug, 2 cocker spaniel, 1 Doberman, 1 pit bull, 1 Australian shepherd) presented for elective ovariohysterectomy and were used in the present study after obtaining written informed owner consent. The sample size was determined by the method described by Charan and Biswas [36] in relation to clinical studies. The mean weight and age with their standard deviation (SD) of these animals was 11.2 ± 6.2 kg and 2.7 ± 1.7 years, respectively. Animals were received and placed in individual accommodation 24 hours before surgery with water and food *ad libitum*. The fasting time was 8 hours of solids and 2 hours of liquids. No animals received any medications prior to surgery and were determined to be clinically healthy based on a physical examination, complete cell blood count, urinalysis, and serum biochemical analysis.

The present study was approved by the Bioethics and animal welfare committee (COBYBA abbreviation in Spanish) of the Veterinary Medicine School, University of the State of Mexico (UAEM).

## Experimental design

Animals were randomly assigned to one of three treatments. Randomisation was performed by number generation (Excel 2010; Microsoft Office). In all treatment groups, analgesic drugs were administered IV before induction of anaesthesia, 30 minutes before the start of surgery. Paracetamol group animals received 15 mg kg$^{-1}$ (Tempra IV, Reckitt Benckiser, México), carprofen group animals received 4 mg kg$^{-1}$ (Rimadyl, Zoetis, México), and meloxicam group animals received 0.2 mg kg$^{-1}$ (Meloxi-Jet NRV, Norvet, México). Postoperatively, the interval to the second dose was measured from the first dose, where paracetamol was administered every 8 hours orally and carprofen and meloxicam were given every 24 hours IV for 48 hours in the same doses, except for meloxicam, which was reduced to 0.1 mg kg$^{-1}$.

## Anaesthesia, surgical procedure, and post-surgical monitoring

All anaesthesia procedures were performed by the same veterinary anaesthetist, who was unaware of the treatment assignments. All animals were aseptically catheterised via the cephalic vein. Anaesthesia was induced with intravenous (IV) propofol (Recofol, Pisa, México) using a titrated dose between 6–8 mg kg$^{-1}$, to allow endotracheal intubation with a cuffed tube. The patient was then connected to an anaesthetic rebreathing circuit with an oxygen flow of 45 mL kg$^{-1}$ minute$^{-1}$ and allowed to breathe spontaneously. Anaesthesia was maintained via the administration of isoflurane (Forane, Baxter Laboratories, USA) vaporised in 100% oxygen with an initial end-tidal isoflurane concentration (ET$_{ISO}$) of 1.3%. This concentration was increased or decreased based on the depth of anaesthesia required for surgery. To accomplish this, the isoflurane vaporiser delivery dial was adjusted to deliver a sufficient concentration based on clinical signs, including absence of the palpebral reflex, absence of jaw tone, and a mean arterial pressure (MAP) between 60 and 90 mmHg. An isotonic fluid solution (0.9% sodium chloride Solution, HT, Pisa Agropecuaria, Mexico) was administered at a flow rate of 10 mL kg$^{-1}$ hour$^{-1}$ immediately after induction through a catheter placed in the cephalic vein.

Five minutes later, a 5 µg kg$^{-1}$ dose of fentanyl citrate (Fenodid, Pisa, México) was administered IV, followed by a continuous rate infusion (CRI) of 5 µg kg$^{-1}$ hour$^{-1}$, using a syringe pump (Module DPS, Orchestra SP, Fresenius Vial Kabi, France), for the duration of the surgery. A 22-gauge catheter was aseptically placed in the dorsal metatarsal artery and attached to a transducer (DTX plus DT 4812; Becton Dickinson Critical Care Systems, Singapore), previously verified against a mercury manometer at 50, 100, and 200 mmHg and zeroed at the level of the manubrium for direct monitoring of arterial blood pressure (systolic [SAP], diastolic [DAP], and MAP). A non-depolarizing neuromuscular blocker, rocuronium bromide (Lufcuren, Pisa, México), was administered IV at 0.6 mg kg$^{-1}$ fifteen minutes before the beginning of surgery and dogs were immediately placed on intermittent mechanical ventilation (Fabius GS, Dräger, Drägerwerk AG & Co., Germany) to maintain an end-tidal CO$_2$ (ETCO$_2$) between 35–45 mmHg (BeneView T5, Mindray Bio-medical Electronics Co., Germany). Neuromuscular function was monitored by acceleromyography (Stimpod NMS450, Xavant technology Ltd., South Africa) through stimulation of the ulnar nerve using a train of four (ToF) and ToF ratio (ToFr) T4-T1. During anaesthesia, heart rate, ETCO$_2$, arterial blood pressure, arterial haemoglobin oxygen saturation (SpO$_2$), and ET$_{ISO}$ were monitored every 5 minutes for the duration of the procedure. A warming blanket (Equator EQ-5000 warming device, Smiths

Medical ASD Inc., USA) was used to maintain each animal's oesophageal body temperature within a physiological range (36–38°C).

An ovariohysterectomy was performed by the same surgeon using a midline surgical approach and a triple haemostat technique. At the end of surgery, CRI fentanyl was discontinued and dogs were allowed to recover in a quiet area. They resumed spontaneous breathing without muscle relaxant reversal and were extubated when their ToFr was greater than 90%. The duration of anaesthesia was recorded from the time of induction to extubation.

### Evaluation of postoperative pain

The same veterinary anaesthetist was also responsible for completing pain assessments before anaesthesia induction (baseline) and at 1, 2, 4, 6, 8, 12, 16, 20, 24, 36, and 48 hours after surgery (postoperative) using the DIVAS and UMPS scoring systems. For the DIVAS, pain was scored using a numerical, 100 mm scoring line in accordance with the level of pain animal perceived and the interaction between the animal and the evaluator, which included palpation of the wound site [37]. For the UMPS, multiple categories of physiological parameters were assessed, including response to palpation, activity, mental state, posture, and vocalisation when it is touched and even if it is intermittent or continuous [38] (S1 Appendix). A technician administered each treatment and the evaluator was blind to animals' group assignments.

Rescue analgesia was administered when a score $\geq$ 40 mm on the DIVAS or $\geq$ 10 points on the UMPS was recorded [38,39]. Tramadol (2 mg kg$^{-1}$, IV) (Tramadol Jet NRV injection; Norvet, México) was used for rescue analgesia [40]. Animals that received rescue analgesia were reported but were not included in subsequent statistical tests.

### Laboratory data

Jugular venous blood samples were collected for complete blood cell counts and serum biochemical analyses, including those of ALT, AST, ALP, direct bilirubin, indirect bilirubin, total bilirubin, creatinine, urea, albumin and glucose at T0 (pre-anaesthesia; baseline), and at 48 and 96 hours postoperatively (T48, T96).

### Statistical analysis

Statistical analyses were performed using Prism 8.1.1 (GraphPad Software, Inc, USA). The Shapiro-Wilk test was used to assess data normality. A repeated measures ANOVA test, followed by a Holm-Sidak multiple post-hoc test, was used to analyse laboratory data and for HR, MAP, temperature, SpO$_2$, ETCO$_2$, and ET$_{ISO}$. Data are reported as mean ± standard deviation (SD). The Friedman non-parametric ANOVA test followed by a Dunn's test was used to analyse postoperative pain, as measured by the DIVAS and UMPS. Medians for this non-parametric data are reported (min, max). Values were considered statistically different when $P < 0.05$.

## Results

Anaesthesia and surgery were uneventful in all dogs. Cardiorespiratory values during anaesthesia were within acceptable normal ranges for all anesthetised patients. However, MAP was significantly lower ($P < 0.001$) in the carprofen group than in the other groups. ETCO$_2$ was also significantly higher ($P < 0.001$) in the paracetamol group than in the other groups, like temperature ($P < 0.01$). The values are shown in Table 1 (S1 Table).

Postoperative pain assessment scores were below established rescue values in all study groups, independent of the scoring system, with the exception of one dog in each of the paracetamol and meloxicam groups and two dogs in the carprofen group that required analgesic

**Table 1. Mean ± SD anaesthesia times, cardiorespiratory variables, temperature, and $ET_{ISO}$ in bitches post-ovariohysterectomy after receiving IV carprofen (4 mg $kg^{-1}$), meloxicam (0.2 mg $kg^{-1}$), or paracetamol (15 mg $kg^{-1}$) before surgery.**

| Parameter | Carprofen group | Meloxicam group | Paracetamol group |
|---|---|---|---|
| Duration of anaesthesia (minutes) | 87 ± 22 | 94 ± 17 | 98 ± 13 |
| HR (beats per minute) | 104 ± 22 | 107 ± 19 | 109 ± 29 |
| MAP (mm Hg) | 71 ± 14 * | 82 ± 17 | 84 ± 16 |
| Esophageal temperature (˚C) | 35.9 ± 0.6 | 35.9 ± 1.0 | 36.3 ± 0.9* |
| $SpO_2$ (%) | 98 ± 2 | 98 ± 1 | 97 ± 1 |
| $ETCO_2$ (mm Hg) | 35 ± 4 | 34 ± 4 | 37 ± 4* |
| $ET_{ISO}$ | 1.21 ± 0.40 | 1.12 ± 0.23 | 1.08 ± 0.14 |

* Significantly different from the other groups (P<0.01)

HR: heart rate

MAP: mean arterial pressure

SpO2: arterial haemoglobin oxygen saturation

ETCO2: end-tidal CO2

$ET_{ISO}$: end-tidal isoflurane concentration

rescue. Animals that received rescue analgesia were reported but were not included in subsequent statistical tests. All groups showed a gradual reduction in pain scores throughout the postoperative period; however, both scoring systems suggested no significant difference between the three treatments in pain relief (*P* = 0.99) at each time of evaluation. The scores obtained at each moment of evaluation during the postoperative period are shown in Figs 1 and 2 (S1 and S2 Figs).

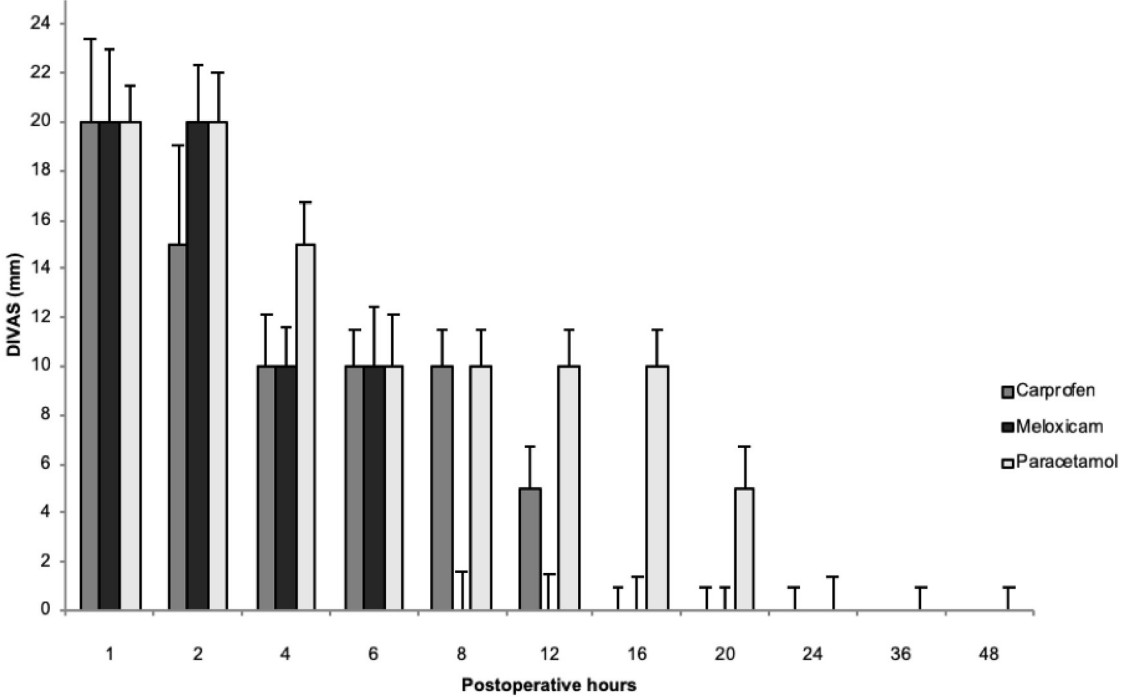

**Fig 1. Postoperative analgesia after ovariohysterectomy in bitches treated with carprofen, meloxicam, or paracetamol via the DIVAS scale (median ± max).** No significant difference was found between the three treatments (P>0.05). Note that measurement values throughout the postoperative period were below 40 mm, the maximum analgesic rescue rating for this system of pain assessment.

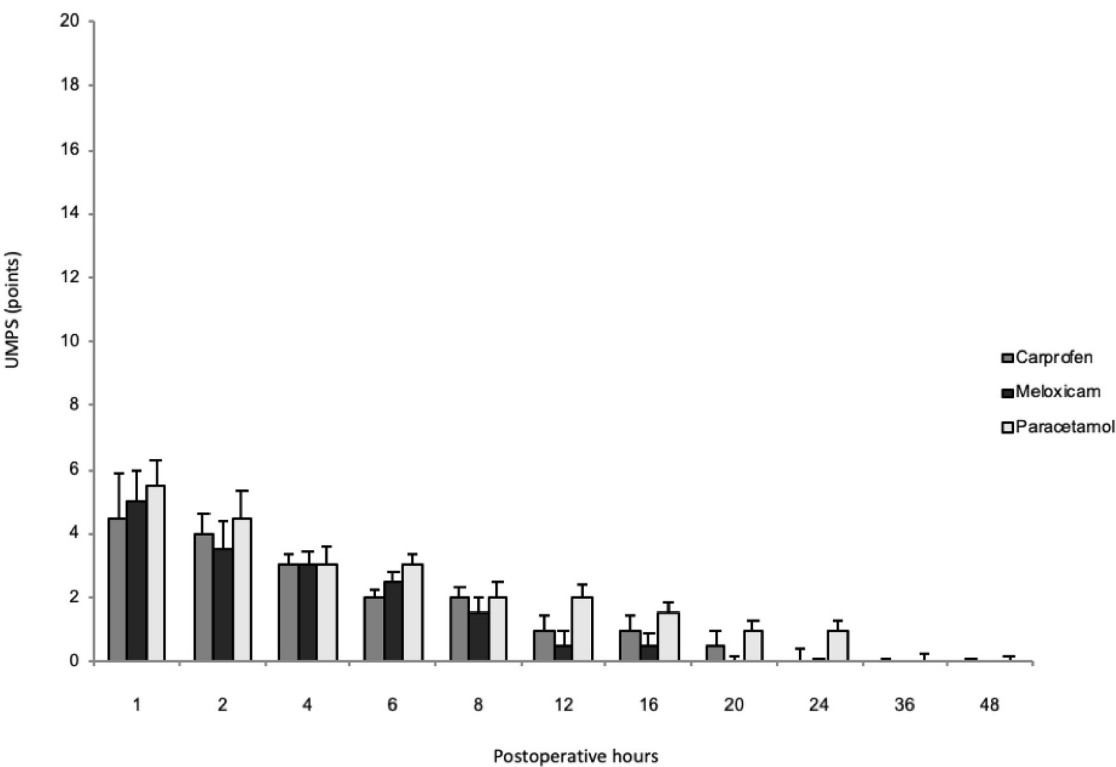

**Fig 2. Postoperative analgesia after ovariohysterectomy in bitches treated with carprofen, meloxicam, or paracetamol through the UMPS scale (median ± max).** No significant difference was found between the three treatments (P>0.05). Note that measurement values throughout the postoperative period were below 10 points, the maximum analgesic rescue rating for this system of pain assessment.

Serum ALT was significantly increased in the paracetamol group and carprofen group at T96 with respect to the baseline. The values of the serum biochemical levels are shown in Table 2 (S2 Table).

## Discussion

Intraoperative pain and nociception in anesthetised animals can be conventionally assessed by detection of haemodynamic reactivity, defined as tachycardia and increased blood pressure, as well as changes in respiratory patterns or movement [41]. However, these modifications are not specific to nociception and may be influenced by anaesthetic agents, as well as by the clinical condition of the animal or the surgery itself [42]. In the present study a neuromuscular blocked was used to generate myo-relaxation and monitored using the ToF, under a correct anaesthetic depth and intraoperative analgesia. Furthermore, no increases in heart rate or blood pressure were detected that required adjustments to the vaporiser setting, likely because animals received a constant rate infusion of fentanyl, which provided adequate analgesia and minimised autonomic responses [41]. In the present study, the dose used (5 μg kg$^{-1}$ loading dose, followed by a CRI of 5 μg kg$^{-1}$ hour$^{-1}$) were similar to those used in other studies, where they have been used previous doses of fentanyl, known to decrease the minimum alveolar concentration of inhalation anaesthetics to prevent motor movement by 22% in dogs [43]. This prior dose has also been shown to decrease heart rate without affecting blood pressure in canine research [44], likely due to a blunting of autonomic responses associated with surgery, as has been found in humans during creation of a surgical incision [41]. Thus, in the present

**Table 2. Serum biochemical levels (mean ± SD) before anaesthesia (T0) and 48 and 96 hours postoperative (T48, T96) in bitches following ovariohysterectomy and IV carprofen (4 mg kg⁻¹), meloxicam (0.2 mg kg⁻¹), or paracetamol (15 mg kg⁻¹) prior to surgery.**

| | Carprofen group | | | Meloxicam group | | | Paracetamol group | | | Canine reference value |
|---|---|---|---|---|---|---|---|---|---|---|
| | T0 | T48 | T96 | T0 | T48 | T96 | T0 | T48 | T96 | |
| ALT | 67 ± 4 | 74 ± 6 | 76 ± 7 * | 67 ± 3 | 73 ± 6 | 74 ± 6 | 66 ± 5 | 72 ± 5 | 80 ± 11 * | 4–73 UI/L |
| AST | 62 ± 12 | 68 ± 9 | 70 ± 9 | 60 ± 6 | 64 ± 9 | 65 ± 8 | 62 ± 9 | 69 ± 11 | 70 ± 18 | 6–70 UI/L |
| ALP | 87 ± 32 | 104 ± 44 | 96 ± 24 | 86 ± 28 | 95 ± 29 | 95 ± 25 | 79 ± 18 | 95 ± 28 | 103 ± 30 | 65–300 UI/L |
| Direct bilirubin | 0.3 ± 0.1 | 0.3 ± 0.1 | 0.4 ± 0.1 | 0.3 ± 0.0 | 0.3 ± 0.1 | 0.3 ± 0.1 | 0.3 ± 0.1 | 0.4 ± 0.1 | 0.4 ± 0.1 | 0.0–0.4 mg/dL |
| Indirect bilirubin | 0.1 ± 0.1 | 0.2 ± 0.0 | 0.1 ± 0.1 | 0.1 ± 0.1 | 0.2 ± 0.1 | 0.1 ± 0.1 | 0.1 ± 0.1 | 0.1 ± 0.1 | 0.2 ± 0.1 | 0.0–0.3 mg/dL |
| Total bilirubin | 0.4 ± 0.1 | 0.5 ± 0.1 | 0.5 ± 0.1 | 0.4 ± 0.1 | 0.5 ± 0.1 | 0.5 ± 0.1 | 0.4 ± 0.1 | 0.5 ± 0.1 | 0.5 ± 0.2 | 0.1–1 mg/dL |
| Urea | 31.2 ± 3.6 | 34.3 ± 4.0 | 34.6 ± 3.4 | 33.6 ± 4.9 | 35.2 ± 4.9 | 35.5 ± 4.2 | 33.5 ± 3.4 | 35.9 ± 4.2 | 36.4 ± 3.9 | 18–40 mg/dL |
| Creatinine | 0.6 ± 0.2 | 0.7 ± 0.1 | 0.7 ± 0.1 | 0.6 ± 0.1 | 0.8 ± 0.2 | 0.8 ± 0.2 | 0.6 ± 0.2 | 0.7 ± 0.3 | 0.7 ± 0.2 | 0.5–1.9 mg/dL |
| Albumin | 3.3 ± 0.4 | 3.3 ± 0.5 | 3.3 ± 0.5 | 3.4 ± 0.5 | 3.2 ± 0.5 | 3.3 ± 0.5 | 3.3 ± 0.5 | 3.5 ± 1.0 | 3.2 ± 0.3 | 2.5–4.4 g/dL |
| Glucose | 98 ± 37 | 99 ± 33 | 99 ± 24 | 85 ± 10 | 83 ± 11 | 88 ± 9 | 83 ± 10 | 84 ± 18 | 88 ± 11 | 60–110 mg/dL |

* Significantly different ($P < 0.01$) with respect to T0

ALT: alanine transaminase

AST: aspartate transaminase

ALP: alkaline phosphatase

experiment, during transoperative cardiorespiratory parameter monitoring, no significant changes were observed and all dogs anesthetised remained within acceptable normal limits, which indicates cardiorespiratory stability during this period.

In recent decades, preventing and controlling postoperative pain has become an important issue in veterinary anaesthesiology, particularly in the surgical care of dogs and cats. In this regard, studies have shown that pain induced by ovariohysterectomy can affect the behaviour of dogs up to 24 hours post-surgery [45,46]. In the present study, however, we did not find significantly altered post-operative behaviour, likely due to the three postoperative pain treatments used (paracetamol, meloxicam, and carprofen). These treatments were equally effective in pain control and led to similar DIVAS and UMPS scores. This is consistent with other studies, where meloxicam and carprofen are effective in the control of postsurgical acute pain in dogs [47–49].

In the postoperative period, pain assessments were performed via the DIVAS and UMPS scales. These two scales are widely considered to be valid tools for the evaluation of post-operative pain in dogs [37,38,47,50]. Both scales are based on behavioural observation and measurement of physiological variables: physiological constants (heart and respiratory rates, rectal temperature), response to palpation, activity, mental state, posture, and vocalisation, which increases specificity and sensitivity, especially relative to baseline animal behaviour [51]. Given these benefits, the DIVAS and UMPS was selected for use in the present study for the postoperative assessment of pain in dogs. We were also able to corroborate pain management across all groups, given that pain scores were below those indicating rescue analgesia use.

NSAIDs provide analgesia and may prevent central sensitisation to painful stimuli caused by persistent stimulation dorsal horn neurons in spinal cord. Despite their desirability, the postsurgical use of NSAIDs is moderate and not universal (as mentioned in the introduction) due to concerns about the possible adverse effects of NSAIDs on renal function. This justifies further medical assessment of the use of these drugs in animals that are in a renal state dependent on prostaglandin [11]. Given this background, hypotension and increased sympathetic tone are common effects of surgery and anaesthesia which may cause renal blood flow changes dependent on prostaglandin E2 and prostacyclin I2 [13]. Because renal protective

prostaglandins are derived primarily from the actions of COX-1, NSAIDs that preferentially target COX-2 may cause less disturbance of renal blood flow and haemodynamics than non-preferential NSAIDs. However, the existing evidence indicates that COX-2 also has a constitutive role in the kidneys [52]. Given this, the present experiment indicates that MAP remained within normal limits because animals' renal filtration rates were not affected during the perioperative period in the carprofen group. However, there were significant decreases, though not sufficient to cause severe hypotension, in this group.

Meloxicam and carprofen have been recognised for their analgesic efficacy for up to 72 hours post-surgery, including after canine ovariohysterectomies [47]. In particular, meloxicam has been used in both acute and chronic pain states. For instance, it has been suggested that meloxicam may control acute postoperative pain for up to 20 hours after a laparotomy, with greater efficacy than that of butorphanol [53]. Carprofen has been widely demonstrated in studies of the perioperative period in dogs [39,48,54] to provide satisfactory analgesia for up to 18 hours post-surgery. Furthermore, premedication, as opposed to postoperative administration, of carprofen provides superior analgesia [12,55]. We observed something similar here, wherein all the study groups exhibited gradual reductions in pain scores during the postoperative period without significant differences between the three treatments used.

The analgesic effects of paracetamol have been evaluated in several studies with, for example, a 33% reduction in post-operative surgical site swelling and a 47% reduction in pain with no adverse effects reported in dogs versus placebo levels after an experimental forelimb surgery [23]. Similarly, no adverse effects occurred in the present study with paracetamol use except for a mild increase in ALT at 96 hours post-surgery, situation that was also observed with the administration of carprofen. This lack of toxicity with controlled use is not unexpected, however, as hepatic toxicity with the use of NSAIDs and paracetamol is mainly due to accidental ingestion, administration of incorrect doses, or overdoses [56,57]. The increase in ALT observed here in the group treated with paracetamol, may be attributed to the metabolite N-acetyl-p-benzoquinone imine (NAPQI), which decreases the capacity of glucuronidation and covalent bonding with many intracellular proteins, thus increasing liver enzymes [15,25,26].

Regarding hepatocellular toxicity induced with carprofen in dogs, an increase in the activity of serum enzymes such as ALT, ALP and AST has been described [58]. In our study, only an increase in ALT was observed at 96 hours of its monitoring, which was not correlated with the induction of the other liver enzymes or with a low serum albumin concentration. In addition, animals treated with carprofen did not show other clinical signs indicative of hepatocellular toxicosis induced by this NSAID, such as loss of appetite, vomiting or jaundice [58]. Some authors report that the prevalence of adverse liver reactions associated with carprofen is probably low, taking into account the small number of cases in which adverse clinical signs or an increase in serum liver enzyme activities have been reported, in relation to widespread use of the medicine [3]. However, the follow-up time in our study was not long enough to reveal most susceptible dogs. In toxicosis of paracetamol, anorexia, weakness, tachypnoea, dyspnoea, cyanosis, icterus, hypothermia, lethargy, and vomiting have been observed in dogs [30] and cats [59]. In contrast to prior reports of such adverse effects, however, we found no changes in clinical laboratory variables or clinical signs 48 hours postoperative in the present study, results which align well with those reported by Serrano *et al.*, [31].

Paracetamol is included in Opioid-free anaesthesia (OFA) protocols, where are often combined with other anaesthetic/analgesic drugs, including medetomidine, ketamine, lidocaine, bupivacaine, carprofen and meloxicam in female dogs. Opioid-free anaesthesia is now more common in human medicine because it has significant implications for clinical care, as opioid dependency is a massive and growing clinical problem in humans. However, in the

future this technique will likely benefit distinct subsets of veterinary patients, including canine ovariohysterectomy patients, since the availability of opioids in veterinary medicine is regulated by legal provisions [60].

Paracetamol has similar analgesic and antipyretic properties to NSAIDs [61]; however, its classification is controversial, since unlike NSAIDs, it has little anti-inflammatory activity and is an inhibitor of the COX-3 isoenzyme, a centrally-located variant of COX-1 in dogs [7]. Some have suggested that paracetamol should be classified as an atypical NSAID or no opioid [62]. Paracetamol can inhibit COX, both centrally and peripherally, when ambient concentrations of peroxides are low. However, under pro-inflammatory conditions, when peroxide concentrations are high, paracetamol is ineffective peripherally and is only active in the brain, where baseline peroxide concentrations are very low [63]. The inhibition of cerebral COX is responsible for the antipyretic effects of paracetamol [64], which is considered the drug of choice in human patients with gastritis, renal disease or with platelet aggregation problems, where the administration of NSAIDs is contraindicated [8].

While the precise mechanism of action underlying paracetamol's efficacy remains unclear, administration of therapeutic doses of this drug in dogs may represent an alternative for the control of acute postoperative pain, as was observed in the present investigation. Furthermore, this drug led to no adverse reactions or cardiorespiratory changes in the parameters assessed in the present study, indicating its safety.

While it offers some significant benefits to the filed, the present study also has some limitations which warrant discussion. Additional studies are needed when considering other forms of nociceptive stimuli given that the present study only examined dogs following elective ovariohysterectomy surgery. It would be interesting to evaluate the analgesic effects of paracetamol following orthopaedic surgery, for example, or for the treatment of chronic pain. Similarly, further investigation of long-term treatments using different doses of paracetamol may be valuable, as the existing veterinary literature suggests large dose ranges (e.g., between 0.2–10 and 1–25 mg kg$^{-1}$, administered intravenously and orally, respectively) [65–67]. A further limitation is that plasma concentrations of paracetamol, meloxicam, and carprofen were not evaluated in the present study. As such, we were unable to correlate plasma concentrations with various cardiorespiratory parameters and analgesic effects in the present study, though future studies should aim to do this with the purpose of avoiding toxicosis and that consequently a greater security of use is provided to paracetamol in dogs. Finally, our use of a small number of animals is an additional limitation, as is the inclusion of only healthy animals in the present study. Future investigations of paracetamol use in dogs of varying ages, sexes, breed, and pathophysiological conditions, even by different routes of administration is required.

In conclusion, the administration of paracetamol provided equivalent analgesic effects to those achieved with meloxicam and carprofen in bitches 48 hours post-ovariohysterectomy in the present study. We found no differences in the adverse effects associated with paracetamol, meloxicam, or carprofen, nor did we observe changes to essential cardiorespiratory parameters with these drugs.

With these results, we could also emphasize the efficacy of the use of NSAIDs for the control of acute post-surgical pain in elective surgery, as proposed in the present study. Although this class of analgesics is also widely used in the treatment of pain located in soft tissues or produced by orthopedic surgery and osteoarthritic pain in dogs. In this way, NSAIDs such as paracetamol, meloxicam and carprofen represent the cornerstone of oral therapy, unless contraindicated by concurrent therapy or by underlying medical conditions that were already discussed in this research paper.

## Supporting information

**S1 Appendix. The University of Melbourne's Pain Scale (UMPS) and the Dynamic Interactive Visual Analog Scale (DIVAS)** [37,38,51].
(DOCX)

**S1 Table. Cardiorespiratory variables (HR, MAP, SpO2, ETCO2), temperature and ET$_{ISO}$ in each individual during the intraoperative in the different study groups.**
(DOCX)

**S2 Table. Serum biochemical levels of ALT, AST, ALP, direct bilirubin, indirect bilirubin, total bilirubin, urea, creatinine, albumin and glucose in each individual during postoperative in the different study groups.**
(DOCX)

**S1 Fig. Scores obtained on the DIVAS scale to determine the degree of pain in each of the individuals of the different study groups for 48 hours postoperatively after performing an elective ovariohysterectomy.**
(DOCX)

**S2 Fig. Scores obtained on the UMPS scale to determine the degree of pain in each of the individuals of the different study groups for 48 hours postoperatively after performing an elective ovariohysterectomy.**
(DOCX)

## Acknowledgments

We wish to thank the Mexican College of Anaesthesiology and Veterinary Analgesia for its support of this study with the necessary equipment and materials.

## Author Contributions

**Conceptualization:** José Antonio Ibancovichi-Camarillo.

**Formal analysis:** Alexander Valverde, Jorge Osorio-Avalos.

**Funding acquisition:** José Antonio Ibancovichi-Camarillo.

**Investigation:** Ismael Hernández-Avalos, José Antonio Ibancovichi-Camarillo, Sergio Recillas-Morales, Desiderio Rodríguez-Velázquez.

**Methodology:** Ismael Hernández-Avalos, Alexander Valverde, Jorge Osorio-Avalos, Desiderio Rodríguez-Velázquez, Agatha Elisa Miranda-Cortés.

**Software:** Jorge Osorio-Avalos.

**Supervision:** Pedro Sánchez-Aparicio.

**Visualization:** Agatha Elisa Miranda-Cortés.

**Writing – original draft:** Ismael Hernández-Avalos, Alexander Valverde, José Antonio Ibancovichi-Camarillo.

**Writing – review & editing:** Alexander Valverde.

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
