## [Decision Letter · Decision Letter 0]

27 Nov 2019

PONE-D-19-26779

Clinical evaluation of postoperative analgesia and serum biochemical changes of paracetamol compared to meloxicam and carprofen in bitches undergoing ovariohysterectomy

PLOS ONE

Dear José A Ibancovichi

Thank you for submitting your manuscript to PLOS ONE. After careful consideration, we feel that it has merit but does not fully meet PLOS ONE’s publication criteria as it currently stands. Therefore, we invite you to submit a revised version of the manuscript that addresses the points raised during the review process.

Please respond the reviewers' comments appropriately.

We would appreciate receiving your revised manuscript by Jan 11 2020 11:59PM. To enhance the reproducibility of your results, we recommend that if applicable you deposit your laboratory protocols in protocols.io, where a protocol can be assigned its own identifier (DOI) such that it can be cited independently in the future. For instructions see: http://journals.plos.org/plosone/s/submission-guidelines#loc-laboratory-protocols

We look forward to receiving your revised manuscript.

Kind regards,

Ehab Farag, MD FRCA FASA

Academic Editor

PLOS ONE

Journal Requirements:

1. At this time, we request that you  please report additional details in your Methods section regarding animal care, as per our editorial guidelines:

(1) Please state the breeds of the dogs used in this study  

(2) Please provide details of animal welfare (e.g., shelter, food, water, environmental enrichment)

(3) Please describe any steps taken to minimize animal suffering and distress, such as by administering anaesthesia  

Thank you for your attention to these requests.

2. Thank you for providing a copy of the criteria used in the University of Melbourne's Pain Scale (UPMS). At this time, we also ask that you please provide a copy of the Dynamic Interactive Visual Analog Scale (DIVAS) socring system if is not under a copyright more restrictive than CC-BY. Please include a copy, in both the original language and English, as Supporting Information to ensure that you have provided sufficient details that others could replicate the analyses.

3. Thank you for stating that consent was obtained from the owners of the female dogs used in this study. At this time, we also ask that you please provide additional details regarding owner consent. Please ensure that you have specified (1) whether consent was informed and (2) what type you obtained (for instance, written or verbal, and if verbal, how it was documented and witnessed).

4. Thank you for including your competing interests statement; " Los autores han declarado que no existen intereses en competencia."

Please update your statement in English

Reviewers' comments:

Reviewer's Responses to Questions

**Comments to the Author**

1. Is the manuscript technically sound, and do the data support the conclusions?

Reviewer #1: Yes

Reviewer #2: Partly

2. Has the statistical analysis been performed appropriately and rigorously? 

Reviewer #1: Yes

Reviewer #2: No

3. Have the authors made all data underlying the findings in their manuscript fully available?

Reviewer #1: Yes

Reviewer #2: Yes

4. Is the manuscript presented in an intelligible fashion and written in standard English?

Reviewer #1: Yes

Reviewer #2: Yes

5. Review Comments to the Author

Reviewer #1: Multimodal analgesia becoming very popular all for the purpose of decreasing pain and avoid the side effect of narcotics.

In conclusion there was no significant difference between the 3 drugs due limited number of the cases in the study and lack of serum concentration of the 3 different drugs.

Reviewer #2: This study is a pain study in female dogs who had hysterectomy and salpingopherectomy. The authors compared the analgesic effect of paracetamol, carboprofen or meloxicam for postoperative pain management and biochemical changes.

It think it is better to replace the biochemical changes with liver and kidney function tests and the hemodynamics. Biochemical changes imply that the blood levels of medications will be measured.

Sample size calculation needs to be included. It is hard to draw conclusions from a study of 10 patients in each group.

I'm not sure why did the study look at the liver and kidney functions while using the therapeutic doses of the medications and expect to have a difference . The only difference that was reported was slight increase of the ALT on day 4 .

The study also, looked at the hemodynamic changes of medications that are known to not cause any significant changes. The only reported a significant change was a decrease of MAP in the carpoprofen group. They found the MAP in the carboprofen group to be 71+/- 14 compared to 82+/- 17 and 84+/- 16 which is still in the acceptable range.

The introduction and the discussion should be shortened.

6. PLOS authors have the option to publish the peer review history of their article (what does this mean?). If published, this will include your full peer review and any attached files.

Reviewer #1: Yes: John Seif MD, MBA

Reviewer #2: No

---

## [Author Response · Author response to Decision Letter 0]

11 Jan 2020

Response to Academic Editor

At this time, we request that you please report additional details in your Methods section regarding animal care, as per our editorial guidelines:

(1) Please state the breeds of the dogs used in this study 

Thanks for your comment the observation has been placed in material and method

Line 172 – 173. (11 mixed breed, 7 poodle, 3 beagle, 2 schnauzer, 2 pug, 2 cocker spaniel, 1 doberman, 1 pit bull, 1 Australian shepherd). 

(2) Please provide details of animal welfare (e.g., shelter, food, water, environmental enrichment)

Thanks for your comment the observation has been placed in material and method

. Line 177 – 179. Animals were received and placed in individual accommodation 24 hours before surgery with water and food ad libitum. The fasting time was 8 hours of solids and 2 hours of liquids.

(3) Please describe any steps taken to minimize animal suffering and distress, such as by administering anesthesia. 

All animals had ownership and are accustomed to handling, catheterization is performed with minimal physical containment.

Subsequently the induction of anesthesia in order to avoid stress. At all times they were monitored (intraoperatively and postoperatively) in order to avoid pain and suffering.

Animals that were observed a score that indicated moderate pain were rescued with the administration of tramadol 2 mg kg intravenously

Thank you for your attention to these requests.

2. Thank you for providing a copy of the criteria used in the University of Melbourne's Pain Scale (UPMS). At this time, we also ask that you please provide a copy of the Dynamic Interactive Visual Analog Scale (DIVAS) scoring system if is not under a copyright more restrictive than CC-BY. Please include a copy, in both the original language and English, as Supporting Information to ensure that you have provided sufficient details that others could replicate the analyses.

Thank you for your comment, the DIVAS scale has been attached to Appendix 1 and the evaluation criteria is described.

3. Thank you for stating that consent was obtained from the owners of the female dogs used in this study. At this time, we also ask that you please provide additional details regarding owner consent. Please ensure that you have specified (1) whether consent was informed and (2) what type you obtained (for instance, written or verbal, and if verbal, how it was documented and witnessed).

Thank you for your comment; the type of consent has been placed.

Line174: presented for elective ovariohysterectomy and were used in the present study after obtaining written informed owner consent.

4. Thank you for including your competing interests’ statement; 

Please update your statement in English

Thank you for your comment

The following statement has been placed on the cover letter:

The authors have stated that there are no competing interests of any kind that interfere with the complete and objective presentation, peer review, editorial decision-making or the publication of research or non-research articles sent to PLOS One, whether financial or non-financial, professional or personal.

Review Comments to the Author

Reviewer #1: Multimodal analgesia becoming very popular all for the purpose of decreasing pain and avoid the side effect of narcotics.

In conclusion there was no significant difference between the 3 drugs due limited number of the cases in the study and lack of serum concentration of the 3 different drugs.

Thank you for your comment

These observations have been included as limitations of the study in the final part of the discussion of this manuscript (Line 375-382)

Reviewer #2: This study is a pain study in female dogs who had hysterectomy and salpingopherectomy. The authors compared the analgesic effect of paracetamol, carprofen or meloxicam for postoperative pain management and biochemical changes.

It think it is better to replace the biochemical changes with liver and kidney function tests and the hemodynamics. Biochemical changes imply that the blood levels of medications will be measured.

Thank you for your comment, the title has been modified for a better understanding of the reader by:

Clinical evaluation of postoperative analgesia, cardiorespiratory parameters and changes in liver and renal function tests of paracetamol compared to meloxicam and carprofen in dogs undergoing ovariohysterectomy

Sample size calculation needs to be included. It is hard to draw conclusions from a study of 10 patients in each group.

Thank you for your comment, this section has been included in materials and methods taking as reference what was cited by Charan and Biswas, 2013, in relation to the determination of the sample size in clinical studies. Line 175-176.

I'm not sure why did the study look at the liver and kidney functions while using the therapeutic doses of the medications and expect to have a difference. The only difference that was reported was slight increase of the ALT on day 4.

Thanks for your comment, however, the use of paracetamol is still rare in dogs, although there are therapeutic doses on the use of paracetamol in dogs, it is also true that there are also several reports that document the liver toxicity of this drug even when therapeutic doses are used.

The study also, looked at the hemodynamic changes of medications that are known to not cause any significant changes. The only reported a significant change was a decrease of MAP in the carpoprofen group. They found the MAP in the carprofen group to be 71+/- 14 compared to 82+/- 17 and 84+/- 16 which is still in the acceptable range.

Thank you for your comment

It is correct all the mean blood pressure values in the different treatments were kept in normal ranges

The introduction and the discussion should be shortened.

Thank you for your comment

The introduction has been shortened

---

## [Editor Report · Decision Letter 1]

23 Jan 2020

Clinical evaluation of postoperative analgesia, cardiorespiratory parameters and changes in liver and renal function tests of paracetamol compared to meloxicam and carprofen in dogs undergoing ovariohysterectomy

PONE-D-19-26779R1

Dear Dr. José A Ibancovichi

We are pleased to inform you that your manuscript has been judged scientifically suitable for publication and will be formally accepted for publication once it complies with all outstanding technical requirements.

With kind regards,

Ehab Farag, MD FRCA FASA

Academic Editor

PLOS ONE
---

## [Editor Report · Acceptance letter]

4 Feb 2020

PONE-D-19-26779R1 

Clinical evaluation of postoperative analgesia, cardiorespiratory parameters and changes in liver and renal function tests of paracetamol compared to meloxicam and carprofen in dogs undergoing ovariohysterectomy 

Dear Dr. Ibancovichi-Camarillo:

I am pleased to inform you that your manuscript has been deemed suitable for publication in PLOS ONE. Congratulations! Your manuscript is now with our production department. 

With kind regards,

on behalf of

Dr. Ehab Farag 

Academic Editor

PLOS ONE